# Dose-Reduced FLA-IDA in Combination with Venetoclax Is an Effective and Safe Salvage Therapy in Relapsed and Refractory Acute Myeloid Leukemia (R/R AML)

**DOI:** 10.3390/cancers16223872

**Published:** 2024-11-19

**Authors:** Martin Schönrock, Piet Sonnemann, Nina Michalowski, Michael Heuser, Felicitas Thol, Francis Ayuketang Ayuk, Christine Wolschke, Evgeny Klyuchnikov, Carsten Bokemeyer, Walter Fiedler, Sophia Cichutek

**Affiliations:** 1Department of Oncology, Hematology and Bone Marrow Transplantation with Section Pneumology, University Medical Center Hamburg-Eppendorf, 20246 Hamburg, Germanyfiedler@uke.de (W.F.); 2Hospital Pharmacy, University Medical Center Hamburg-Eppendorf, 20246 Hamburg, Germany; 3Department of Hematology, Hemostasis, Oncology and Stem Cell Transplantation, Hannover Medical School, 30625 Hannover, Germany; 4Department of Stem Cell Transplantation, University Medical Center Hamburg-Eppendorf, 20246 Hamburg, Germany

**Keywords:** acute myeloid leukemia, relapsed/refractory, FLAG-IDA, venetoclax

## Abstract

In this single-center study, we aimed to assess safety and efficacy in patients undergoing a dose-reduced, 4-day salvage regimen of FLA-VIDA in comparison to full-dose FLA-VIDA, and FLA-IDA alone, to reduce the duration of aplasia while maintaining clinical efficacy. The addition of venetoclax to FLA-IDA improves EFS, without increasing toxicity or prolonging cytopenia; further dose-reduced FLA-VIDA allows for a shorter duration of aplasia compared to the full-dose regimen, without compromising response rates.

## 1. Introduction

Intensive chemotherapy regimens for acute myeloid leukemia (AML) have improved outcomes for patients, and the majority of patients with newly diagnosed AML achieve complete remission (CR). However, 30–40% of patients relapse. This is especially evident for patients with an adverse ELN category or an older age [1,2,3]. Primary treatment failure, which accounts for 10–40% of AML patients, represents a significant challenge in AML treatment. Despite numerous innovations in therapy such as the CD33-taregting antibody-drug-conjugate gemtuzumab ozogamicin (GO) or targeted therapies with FLT3 inhibitors and IDH inhibitors, the prognosis for relapsed or refractory (R/R) AML patients remains limited due to a relatively low response rate to salvage therapy and poor overall survival (OS) [4,5,6]. In larger cohort studies of patients with R/R AML treated with re-induction chemotherapy followed by allogeneic hematopoietic stem cell transplantation (AHSCT), the 5-year overall survival for all patients was only 15–25% [7]. So far, AHSCT remains the only curative therapeutic approach for this patient population. Longer survival was observed in patients with R/R AML who achieved CR through salvage chemotherapy compared to patients not in remission before undergoing AHSCT [8]. This outlines the importance of an effective salvage chemotherapy in R/R AML. No multi-center, prospective, randomized, controlled trial investigating the efficacy of various salvage chemotherapies has been conducted, leading to the absence of a defined standard regimen.

R/R AML is commonly treated with intensive salvage chemotherapy comprising a high-dose cytarabine backbone in combination with an anthracycline, including fludarabine, cytarabine, granulocyte colony stimulating factor with idarubicin (FLAG-IDA), mitoxantrone, etoposide, cytarabine (MEC), cladribine, cytarabine, granulocyte colony stimulating factor (CLAG) or high-dose cytarabine and mitoxantrone (HAM) in combination with GO [2,4,5,9]. Those salvage regimens are effective treatment options for fit patients with R/R AML, achieving CR rates, including CR with incomplete hematologic recovery (CRi) of about 50% [3,6,10]. However, most of these salvage therapies fail to obtain substantial duration of CR or significant OS rates [11].

The common adverse events in R/R AML patients treated with FLAG-IDA, MEC, CLAG or GO-HAM salvage regimens were hematological toxicity and infections due to a median time until neutrophil recovery of 27 days (range 15–46) [12]. As infectious complications constitute the most common cause of morbidity among neutropenic AML patients undergoing intensive chemotherapy, there is a significant need for improving the safety and tolerability of salvage regimens [13,14,15,16]. One possibility is to dose-reduce currently established regimens. In our institution, we implemented a FLA-IDA regimen consisting of only 4 days of chemotherapy to reduce toxicity with maintained efficacy.

A novel approach to FLAG-IDA salvage therapy involves the addition of the BCL-2 inhibitor venetoclax, currently approved only for the first-line treatment of unfit patients [17,18]. Venetoclax combined with FLAG-IDA demonstrated favorable response rates in newly diagnosed (ND) and R/R AML patients, with overall response rates (ORR) of 97% and 70% in the Phase IIA (ND, *n* = 29) and Phase IIB (R/R, *n* = 39) trial, respectively. Median OS in R/R AML patients was 13 months (95% CI, 7-not reached). However, this trial by DiNardo et al. was associated with a high rate of grade 3–4 neutropenia-related infectious complications with overall rates of blood stream infections, pneumonia, and sepsis in 35%, 28%, and 12%, respectively, and resulted in both venetoclax and cytarabine dose reductions. The median time to full hematological recovery was 37 days (neutrophil count ≥ 0.5/nL and platelet count ≥ 100/nL) [19]. Therefore, the administration of FLAG-IDA in combination with venetoclax raised a significant safety concern regarding a notable 30-day mortality rate of 12% [20]. In this regard, attempts have been made to shorten the duration of chemotherapy in addition to venetoclax. In the CAVEAT study, the “5 + 2” regimen consisting of 5 days of cytarabine (100 mg/m^2^), 2 days of idarubicin (12 mg/m^2^), plus 7 days of venetoclax was employed in elderly fit patients with AML [21]. Further, in a phase II trial the “2 + 6” regimen with 2 days of daunorubicin and 6 days of cytarabine in combination with venetoclax led to deep responses in 42 AML patients, including CR and MRD negativity, supporting the concept of reduced chemotherapy in combination with venetoclax [22].

In this single-center study, we aimed to assess the safety and efficacy in patients undergoing a dose-reduced, 4-day salvage regimen of fludarabine, cytarabine, idarubicin in combination with venetoclax (FLA-VIDA) to reduce the duration of aplasia while maintaining clinical efficacy. This was retrospectively compared with the same regimen without venetoclax.

## 2. Materials and Methods

### 2.1. Study Design and Population

In this retrospective single-center analysis, we evaluated the efficacy and safety of a dose-reduced FLA-IDA regimen (fludarabine 30 mg/m^2^ day 1–4, cytarabine 2000 mg/m^2^ day 1–4, idarubicin 10 mg/m^2^ day 1 + 4) in combination with venetoclax (day 1 100 mg, day 2 200 mg, day 3–14 400 mg) in R/R AML patients. Venetoclax was administered orally, and the dose was adjusted to 50 mg or 100 mg daily when co-administered with strong and moderate CYP3 A4 inhibitors, respectively, based on the results of the VIALE-A study [23]. All included patients had previously received 1 first-line therapy, which included induction and consolidation cycles or AHSCT after initial induction therapy, depending on the ELN risk classification. The majority of all patients received standard induction therapy with 7 + 3. In total, 18 patients received CPX-351, of which 3 patients were in the FLA-IDA cohort and 15 were in the FLA-VIDA cohort. All patients received anti-infective prophylaxis, consisting of antibacterial, antiviral, and antifungal agents. No standard granulocyte colony-stimulating factor (G-CSF) was administered, and G-CSF was only used when severe infectious complications occurred. Patients who received previous salvage treatment for current R/R AML were excluded from analysis. Patients who showed a response received an allogeneic stem cell transplant (AHSCT) after one course of salvage therapy. In the event of a non-response to salvage therapy with FLA-IDA/FLA-VIDA, a further attempt at salvage therapy was made if the clinical condition was sufficient, whereby two thirds of the patients could subsequently also undergo AHSCT. Alternatively, palliative therapy was carried out. All patients were treated at the University Hospital Hamburg-Eppendorf, Hamburg, Germany and were included in the UKE AML registry after giving their written consent for data acquisition and analysis after pseudonymization. The retrospective data collection and analysis was performed in accordance with local legal requirements (§12 Hamburgisches Krankenhausgesetz) and approved by the Ethics Committee of the Medical Council of Hamburg (vote number: 2024-300422-WF). The use of venetoclax in combination with FLA-IDA is currently off-label, all patients had given written informed consent.

### 2.2. Safety and Efficacy Assessment

The primary objectives were to explore safety and tolerability of dose-reduced FLA-IDA and FLA-VIDA and to evaluate the overall response rate (ORR) in patients with R/R AML. Secondary objectives were for the assessment of survival outcomes including overall survival (OS; time from treatment initiation to death of any cause) and event-free survival (EFS; time from treatment initiation until death of any cause, refractory disease or relapse, whichever occurred first). Non-responders were considered as progressing on the day of response assessment.

The ORR was defined by European LeukemiaNet (ELN) 2022 criteria and comprised complete remission (CR, defined as bone marrow blasts < 5%; absence of circulating blasts; neutrophil count ≥ 1/nL and platelet count ≥ 100/nL; ≤2 weeks after response assessment), complete remission with incomplete blood count recovery (Cri, all CR criteria except for residual neutropenia < 1/nL or thrombocytopenia < 100), and morphologic leukemia free state (MLFS, defined as less than 5% bone marrow blasts without hematological recovery [24].

Non-hematologic toxicity was evaluated according to the National Cancer Institute Common Terminology Criteria for Adverse Events (CTCAE v5.0). Cell count recovery was assessed only for patients who achieved response (CR, Cri). Thrombocyte time to recovery was defined as the time between the start of the treatment and first thrombocyte count > 100/nL; neutrophil time to recovery cut-off was defined as neutrophil count > 1/nL. The standard operating procedures in regard to anti-infective prophylaxis and treatment did not change between the FLA-IDA and FLA-VIDA cohorts and included antibiotic and antifungal prophylaxis.

### 2.3. Genetic and Molecular Analysis

The ELN 2022 recommendation for genetic risk classification at initial diagnosis was used to stratify patients into favorable, intermediate, or adverse risk [24]. Genetic and molecular analysis of all patients was performed by NGS in Hannover, Germany by M. Heuser and F. Thol using peripheral blood or bone marrow samples before start of chemotherapy at the time of initial diagnosis. Measurable residual disease (MRD) assessment by quantitative PCR or NGS was performed for 24 patients in the FLA-VIDA group and 6 patients receiving FLA-IDA in Hannover, Germany by M. Heuser and F. Thol. Thresholds for MRD negativity were selected in accordance with the ELN recommendations for diagnostics from 2021 [25]. Negativity in NGS-MRD was defined as <0.2% variant allele frequency with a limit of detection of 0.01%.

### 2.4. Statistical Analysis

Statistical analysis was performed in R version 4.3.2, R Core Team, Vienna, Austria [26]. Baseline characteristics were analyzed with descriptive statistics. Group differences were evaluated using Fisher’s exact test. OS and EFS analysis were performed using the Kaplan–Meier method and log-rank testing. Missing data points were omitted from the analysis, as indicated.

## 3. Results

### 3.1. Patient Characteristics

A total of 89 R/R AML patients were treated between 2011 and 2023, with a total of 38 patients receiving dose-reduced FLA-IDA and 51 patients receiving concomitant venetoclax (FLA-VIDA) on days 1–14. The patients were treated with FLA-IDA between 2011 and 2019. From 2019 onwards, venetoclax was added to the FLA-IDA regimen (FLA-VIDA). The follow-up time in the FLA-IDA cohort was 1560 (95% CI 1387 to NR) days and that of the FLA-VIDA cohort 558 (95% CI 387–812) days. The median age was 55 years (19–75 years) in the FLA-VIDA group and 60 years (30–76 years) in the FLA-IDA group. AML subtypes were comparable, with 86.3% and 78.9% de novo AML in the FLA-VIDA vs. control group, respectively, 9.8% vs. 13.2% AML with a prior history of MDS or MPN, and 3.9% vs. 7.9% therapy-related AML (tAML: defined as AML occurring post cytotoxic therapy). Nearly all patients received intensive chemotherapy prior to salvage therapy. Baseline characteristics are presented in Table 1.

There was a significantly higher proportion of primary refractory AML patients in the FLA-VIDA group (66.7% vs. 39.5%, *p* = 0.020). As a result, the rate of prior AHSCT in the FLA-IDA cohort was higher at 28.9% compared to 7.8% in the FLA-VIDA cohort. Over time, the implemented NGS panels have been expanded to include more mutational analyses. At the same time, mutations that are associated with an adverse prognosis were used for stratification of AML patients into risk groups according to the 2022 ELN classification. For 49 patients in the FLA-VIDA cohort and 20 patients in the FLA-IDA cohort, the molecular genetic analyses required for the 2022 ELN risk stratification were available. Patients who were stratified and treated according to previous ELN classification criteria were censored for comparison between the ELN risk groups. Due to the high rate of primary resistant patients, higher rates of prognostically unfavorable mutations such as RUNX1, ASXL1, and SRSF2 were found in this cohort. Consequently, a significantly higher proportion of ELN adverse risk patients were in the FLA-VIDA group (63.3% vs. 15.0%, *p* < 0.001) and a lower proportion of ELN intermediate risk patients in the FLA-VIDA group (12.2% vs. 55.0%, *p* < 0.001). ELN favorable risk patients were found to be comparable between groups (26.5% vs. 30.0%). In conclusion, the FLA-VIDA cohort had a considerably higher risk profile.

### 3.2. Response

A total of 86 patients were available for the response assessment (36 in the FLA-IDA and 50 patients in the FLA-VIDA cohort). The time until the first response analysis was 24.45 (13–41) days in the FLA-IDA group and 24.49 (14–36) days in the FLA-VIDA group. ORR was significantly higher in the FLA-VIDA group with 74.5% of patients achieving a response (CR; Cri or MLFS) vs. 47.3% in the group receiving FLA-IDA only (*p* = 0.032) (Figure 1). The rate of CR/Cri was 60.0% for FLA-VIDA and 38.8% for FLA-IDA. Exploratory subgroup analysis showed a trend favoring FLA-VIDA in all investigated subgroups (Figure 2). FLA-VIDA significantly improved ORR for patients with AML with a prior history of MDS or MPN, patients >60 years at the start of treatment, and female patients. Relapsed and refractory AML patients benefited equally from the addition of venetoclax compared to the FLA-IDA group. In the prognostically less favorable group of primary refractory patients, venetoclax was able to improve the ORR to 74.0% compared to FLA-IDA alone, with 47.3%. Interestingly, patients who had prior AHSCT also benefitted from the addition of venetoclax. From the group of TP53 mutated patients, one of a total of two patients treated with FLA-IDA and five of a total of eleven patients treated with FLA-VIDA achieved remission.

MRD response assessment was available for twenty-four patients in the FLA-VIDA group and six patients in the FLA-IDA group. A total of 45.8% (*n* = 11/24) of FLA-VIDA treated patients achieved MRD negativity measured by qPCR or NGS. In the FLA-IDA group, 16.7% (*n* = 1/6) of patients achieved MRD negativity.

### 3.3. Allogeneic Hematopoietic Cell Transplantation

In total, 72 of 89 (80.9%) R/R AML patients received AHSCT following salvage therapy (Appendix A). In the FLA-VIDA cohort 44 patients (86.3%) transitioned to AHSCT, while only 28 patients (73.7%) receiving FLA-IDA proceeded to AHSCT. Among responding patients, 92.1% (*n* = 35/38) of the FLA-VIDA cohort, and 83.3% (*n* = 15/18) of the FLA-IDA cohort proceeded to AHSCT (no statistically significant difference).

### 3.4. Survival

Addition of venetoclax to FLA-IDA significantly improved EFS (Figure 3). The median EFS was 594 days in the FLA-VIDA group compared to 39.5 days in the FLA-IDA group (logrank *p* = 0.01). The 1-year EFS was 52.93% in the FLA-VIDA group compared to 31.58% in the FLA-IDA group. The median OS was 594 days (95CI 297 to NR) in the FLA-VIDA group and 499 days (CI 230-NR) in the FLA-IDA group (logrank test *p* = 0.6) (Figure 4). The 1-year OS was 61.45% in the FLA-VIDA group compared to 56.61% in the FLA-IDA group.

### 3.5. Safety, Toxicity and Mortality

Adverse events (AE) were common with severe hematological toxicity occurring in all patients following salvage therapy until AHSCT. The most common Aes were severe pancytopenia (100% in both groups), febrile neutropenia (88.2% vs. 92.1%), and nausea (15.8% vs. 15.7%) for FLA-VIDA and FLA-IDA treated patients. Admittance to the intensive care unit was necessary in 13.7% of FLA-VIDA patients and 15.8% of FLA-IDA, primarily due to respiratory failure or sepsis. Bloodstream infections occurred in 32.0% of FLA-VIDA patients and 54.4% of FLA-IDA patients, with Gram-positive bacteremia occurring more frequently (26.0% vs. 40.5%). Elevation of liver enzymes (15.8% vs. 2.0%) and acute kidney injury > KDIGO 2 (10.5% vs. 2.0%) was more frequent with FLA-IDA, with complete recovery in most cases. Gastrointestinal adverse events such as nausea and diarrhea were overall moderate and manageable (Table 2). Tumor lysis syndrome was rare in both groups (7.9% vs. 11.8%) with no statistically significant difference. Overall, three grade 5 adverse events leading to death were noted in the FLA-IDA group with two deaths due to intracranial hemorrhage and one due to acute heart failure. No deaths directly associated with the treatment were noted in the FLA-VIDA group. The detected deaths occurred in the context of sepsis and hemorrhage during cytopenia or AHSCT. The 30-day mortality with FLA-IDA and FLA-VIDA was 13.16% and 5.88%, respectively, and the 60-day mortality was 18.42% and 7.84%. The causes of death with FLA-IDA were two hemorrhages, three severe infections/sepsis, one AML progression, and one heart failure. The causes of death with FLA-VIDA were two severe infections/sepsis and one stroke.

The median thrombocyte and neutrophil recovery were not significantly increased in the venetoclax group. The median time to ANC > 0.5/nL was 25 days in the venetoclax group and 27.5 days in the FLA-IDA groups. The median time to ANC > 1.0/nL was 29.0 days for FLA-VIDA and 32.5 days in the FLA-IDA group. The platelet recovery to PLT > 50/nL was 38.5 days and 38.5 days in the FLA-VIDA and FLA-IDA groups, respectively. Recovery to PLT > 100/nL was 40.5 days for FLA-VIDA and 40 days for FLA-IDA (Figure 5). Two patients received AHSCT during aplasia after confirmation of MLFS on day 21 and 22 after the start of salvage therapy, respectively.

## 4. Discussion

In this retrospective study, our objective was to evaluate the safety and efficacy of a dose-reduced FLA-VIDA in comparison to FLA-IDA regimen in R/R AML patients, within a real-world, single-center setting.

With 47.3%, the ORR of the dose-reduced FLA-IDA protocol is comparable to the ORR of the standard full-dose FLA-IDA protocol reported in the literature with an ORR of around 50% [10,12]. Further, the dose-reduced FLA-IDA showed comparable regeneration times for neutrophils with 27 days (95% CI 24–28, neutrophil count ≥ b0.5/nL) and for platelets with 38.5 days (95% CI 26-NR, platelet count ≥ 50/nL), when compared to full-dose FLA-IDA protocols [12,27].

The combination of dose-reduced FLA-IDA and venetoclax resulted in a significantly improved ORR of 74.5% compared to 47.3% with FLA-IDA alone. In a phase IIB trial (*n* = 39), venetoclax combined with standard FLAG-IDA demonstrated favorable response rates in R/R AML patients with an ORR of 70% [19]. In a single-center retrospective study of patients with R/R AML, 37 and 81 patients received FLA-IDA with or without venetoclax, respectively. FLA-VIDA granted high response rates with an ORR of 78% vs. 47% [27]. In conclusion, the ORR with dose-reduced FLA-VIDA were comparable with previously published data from patient cohorts treated with full-dose FLA-VIDA with ORR ranging between 70 and 78% [19,27]. In general, the advantage in response rate of FLA-VIDA is maintained over all groups. Interestingly, in refractory patients, pretreatment with the standard regimen “7 + 3” or with CPX-351 did not influence the ORR.

The addition of venetoclax to FLA-IDA primarily resulted in a higher proportion of CR despite a significantly larger proportion of patients with adverse risks in our FLA-VIDA cohort. Furthermore, there were more patients with primary refractory disease in the FLA-VIDA cohort compared to the FLA-IDA cohort. With better-balanced groups, an even clearer advantage of the FLA-VIDA could possibly be expected.

MRD data, determined by qPCR or NGS, were only available for a small subset of patients (FLA-VIDA, *n* = 24; FLA-IDA, *n* = 6). MRD negativity was achieved after a single treatment cycle in 45.8% of patients in the FLA-VIDA cohort. In our study MRD negativity in patients receiving FLA-VIDA did not improve OS, thus limiting the prognostic significance for this patient group. In a retrospective study by Shashwar et al. MRD negativity after FLA-VIDA salvage therapy of R/R AML also did not result in an improved OS [27].

In our study, the combination of FLA-VIDA showed a significantly improved EFS. In particular, primary refractory patients showed a significantly improved EFS. The median OS in R/R AML patients receiving FLA-VIDA was 594 days (95CI 297 to NR) compared to 499 days (95CI 230 to NR) with FLA-IDA alone. A multi-center retrospective cohort study of 25 R/R AML patients receiving standard FLAG-IDA in combination with venetoclax demonstrated an OS at 12 months of 50% (95CI 31 to 69) [20]. In a single-center retrospective study of patients with R/R AML by Shahswar et al., 37 and 81 patients received FLA-IDA with or without venetoclax, respectively. In addition, high response rates with an ORR of 78% for FLA-VIDA vs. 47% for FLA-IDA only, the median EFS and OS were not improved in FLA-VIDA treated patients [27]. However, most results rely on small, single-center retrospective studies. In conclusion, the dose-reduced FLA-VIDA protocol achieved comparable EFS and OS when compared to full-dose FLA-VIDA protocols reported in the literature [10,12,27].

Although the addition of venetoclax significantly improved response rates and EFS, this did not translate into a significant impact on OS. This is consistent with previously published studies in which improved EFS did not lead to an increase in OS. In total, 66% of non-responders in the FLA-VIDA group and 72.2% in the FLA-IDA group proceeded to salvage treatment and AHSCT, which impacts overall survival. In our subgroup analysis, ELN intermediate and adverse risk AML patients showed a non-significant trend toward a better OS when receiving FLA-VIDA. In a previous study, a benefit in terms of OS was also only seen in the context of newly diagnosed adverse risk AML [19]. Despite improved response rates and EFS, AHSCT remains the most important factor to ensure long-term survival in R/R AML.

Remarkably, the addition of venetoclax to the dose-reduced FLA-IDA in our study did not result in an increased duration of neutropenia or thrombopenia compared to FLA-IDA alone, opposed to regeneration times described regarding the combination of venetoclax with full-dose FLA-IDA [19,27]. In addition to comparable hematologic regeneration times, the combination of FLA-IDA and venetoclax did not lead to a significant increase in higher grade adverse events. The FLA-VIDA group did not show an increased rate of complications such as severe infections, sepsis, admissions to the intensive care unit or the occurrence of tumor lysis syndrome. Due to the encouraging improvement in ORR and EFS in R/R AML patients, will venetoclax containing regimen be moving to first line therapy? Chua et al. showed an impressive CR rate of 72% in elderly fit AML patients with a reduced chemotherapy regimen “5 + 2” in the CAVEAT study [21]. In a phase II trial 42 patients with de novo AML were treated with a “2 + 6” regimen in combination with venetoclax [22]. Here, Suo et al. could show an ORR of 92.9% with a high rate of MRD negativity. The Acute Myeloid Leukemia Study Group (AMLSG) has initiated a placebo-controlled phase III study of induction and consolidation chemotherapy with venetoclax in patients with newly diagnosed AML or MDS/AML (AMLSG-31-19; ClinicalTrials.gov/NCT04628026) [28]. When these results become available, the impact of venetoclax-based intensive treatment regimen will become clearer. In the phase I/II RELAX trial of the Study Alliance Leukemia (SAL), R/R AML patients receiving high-dose cytarabine, mitoxantrone, and venetoclax (HAM-Ven) as a salvage therapy showed promising response rates, with CR/CRi in 31/38 patients (81.6%) [29,30]. The limitations of this study are the retrospective nature, non-randomized cohorts, and a shorter follow-up period of the FLA-VIDA cohort. We note that there is some imbalance in regard to baseline characteristics, which affects the interpretation of the results. In the historical FLA-IDA cohort, complete NGS diagnostics were available for only 20 out of 38 patients, while in the more recently treated FLA-VIDA cohort, it was available for 37 out of 51 patients. Thus, subgroup analyses, such as IDH1/IDH2 mutational status, while consistent with previous studies, should be considered exploratory. Further studies with larger cohorts are needed to assess which populations have the greatest benefit of this treatment. Furthermore, the rate of previously transplanted patients was higher in the FLA-IDA group; however, excluding these patients did not significantly impact results.

## 5. Conclusions

In conclusion, our findings indicate that dose-reduced FLA-VIDA is an effective and safe intensive salvage therapy for R/R AML, especially as bridging approach toward AHSCT. FLA-VIDA showed high overall response rates and a significantly improved EFS in this difficult-to-treat AML patient population.

## Figures and Tables

**Figure 1 cancers-16-03872-f001:**
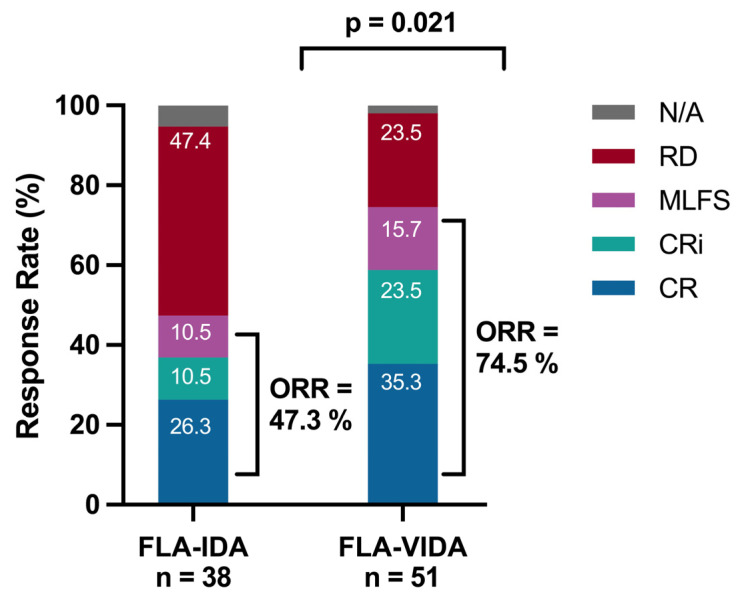
Comparison of remission rates and overall response rates (ORR) in patients treated with FLA-VIDA vs. FLA-IDA regimen. The response analysis for overall response rate was performed with Fisher’s exact test. N/A: Remission status not evaluable; RD: Refractory disease; ORR: Overall response rate; CR: Complete remission; CRi: Complete remission with incomplete hematological recovery; MLFS: Morphologic leukemia free state.; FLA-VIDA: fludarabine, cytarabine, idarubicin and venetoclax; FLA-IDA: fludarabine, cytarabine, idarubicin without venetoclax.

**Figure 2 cancers-16-03872-f002:**
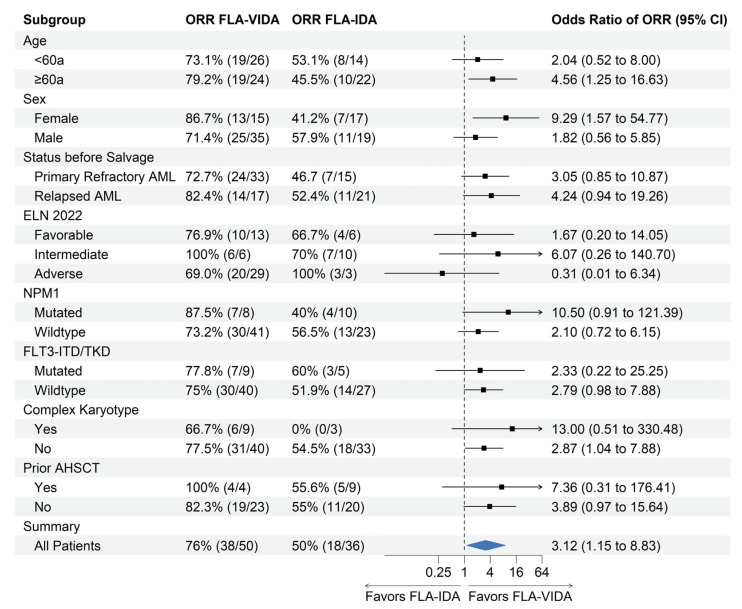
Subgroup analysis for overall response in patients treated with FLA-VIDA vs. FLA-IDA regimen. Odds ratio and confidence intervals calculated according to Altmann 1991. For zero event cells Haldane Correction was performed. Patients without remission evaluation were excluded from analysis (*n* = 1 in FLA-VIDA; *n* = 2 in FLA-IDA cohort). Patients without complete NGS were excluded from ELN stratification. Subgroup analysis for IDH1/2 and TP53 omitted due to small sample size. ORR: Overall response rate (CR/CRi/MLFS); FLA-VIDA: fludarabine, cytarabine, idarubicin and venetoclax; FLA-IDA: fludarabine, cytarabine, idarubicin without venetoclax; ELN 2022: European Leukemia Net 2022; AHSCT: allogeneic hematopoietic stem cell transplantation.

**Figure 3 cancers-16-03872-f003:**
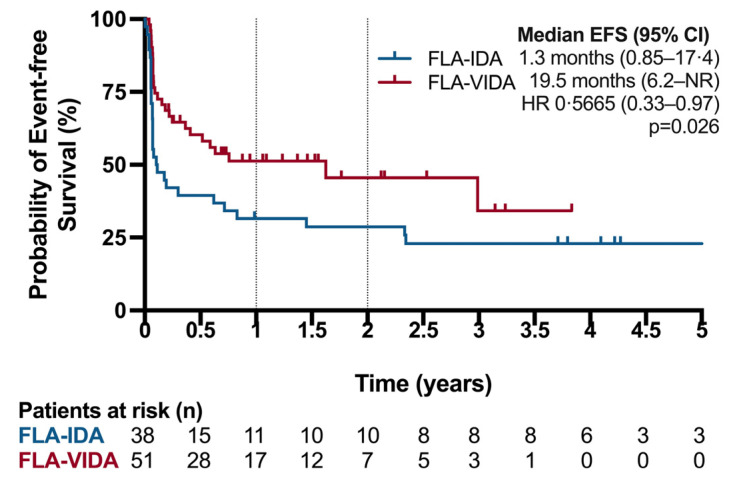
Event-free survival (EFS) in patients treated with FLA-VIDA vs. FLA-IDA regimen. Events were defined as death of any cause, relapse, or failure to achieve remission/refractory disease. For patients failing to achieve remission time of event was defined as date of first response assessment. HR: Hazard ratio.

**Figure 4 cancers-16-03872-f004:**
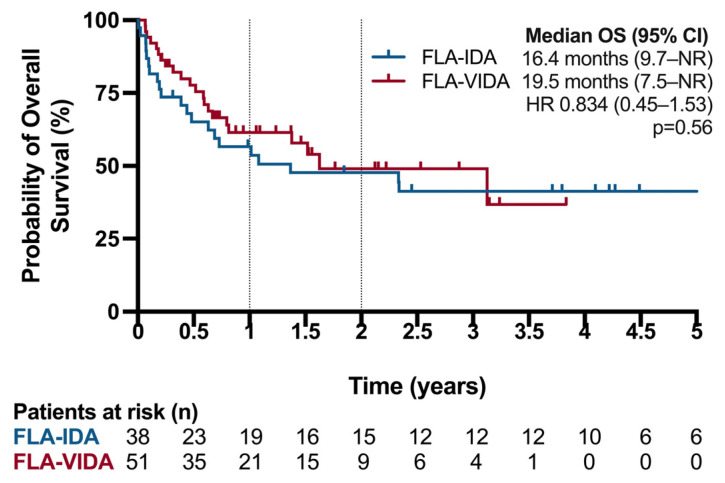
Overall survival (OS) in patients treated with FLA-VIDA vs. FLA-IDA regimen. HR: Hazard ratio.

**Figure 5 cancers-16-03872-f005:**
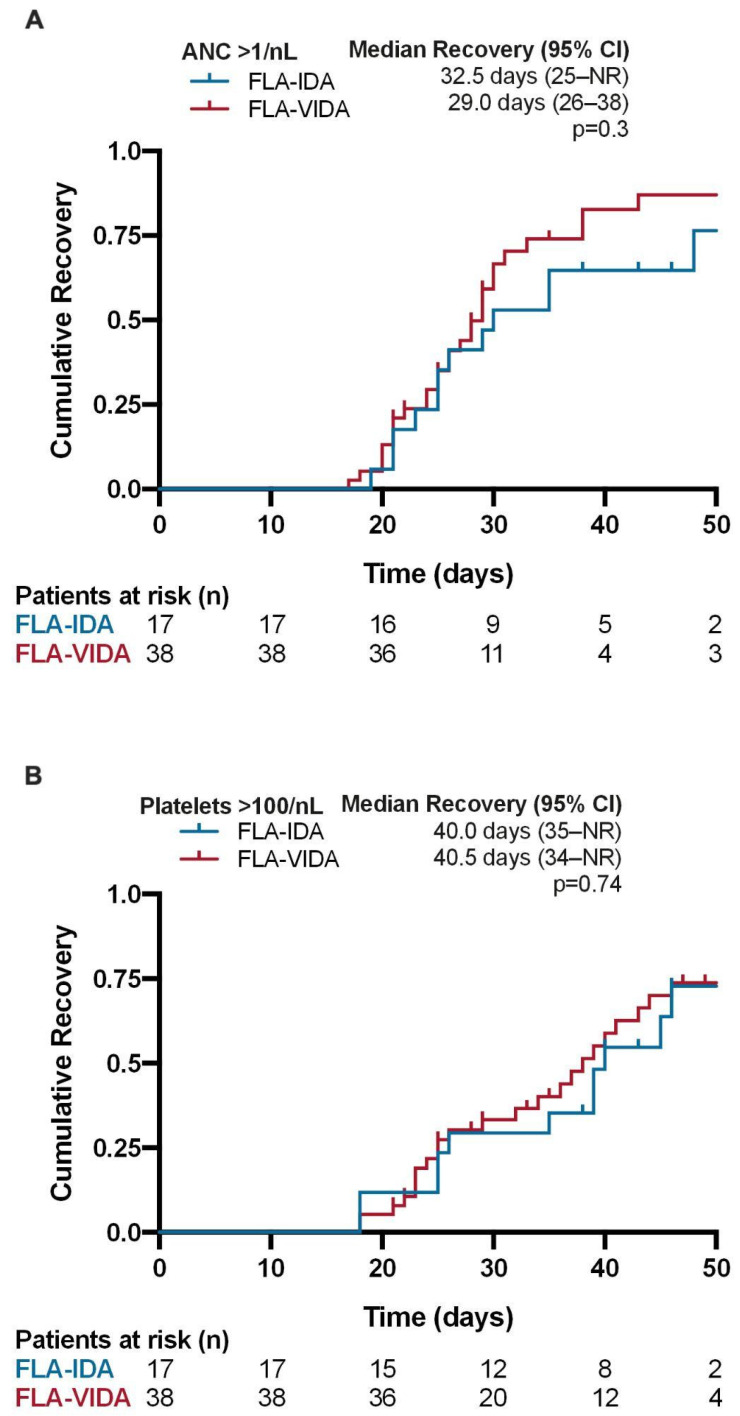
Time to neutrophil and platelet recovery in responding patients treated with FLA-VIDA vs. FLA-IDA regimen. (**A**) Absolute neutrophil count (ANC) recovery > 1/nL. (**B**) Platelet recovery (PLT) > 100/nL. Only patients achieving CR, Cri, or MLFS were included in the analysis. Patients who did not achieve complete recovery within 60 days of administration of FLA-VIDA/FLA-IDA were included in the analysis. Patients who proceeded to transplantation prior to hematological recovery were censored. *p*-values were calculated via log-rank test.

**Table 1 cancers-16-03872-t001:** Baseline characteristics of patients treated with FLA-VIDA vs. FLA-IDA regimen. Patients with unknown mutational status were excluded from calculation of percentages; total number of patients included in calculations as listed. If mutational profiles were incomplete for establishment of ELN risk stratification, patients were excluded. *n*: number, AML: acute myeloid leukemia, FLA-VIDA: fludarabine, cytarabine, idarubicin and venetoclax, FLA-IDA: fludarabine, cytarabine, idarubicin without venetoclax; ELN 2022: European Leukemia Net 2022; AHSCT: allogeneic hematopoietic stem cell transplantation, HMA: hypomethylating agents.

Baseline Characteristics		
	FLA-VIDA [*n*, (%)]	FLA-IDA [*n*, (%)]
Total number of patients	51	38
Sex (Female)	15 (29.4)	18 (47.4)
Age at start of salvage treatment	55.12 (15.33)	60.29 (11.24)
AML type		
De novo AML	44 (86.3)	30 (78.9)
AML with prior MDS/MPN	5 (9.8)	5 (13.2)
Treatment-associated AML	2 (3.9)	3 (7.9)
AML status before salvage		
Relapsed AML	17 (33.3)	23 (60.5)
Primary refractory AML	34 (66.7)	15 (39.5)
Induction treatment		
7 + 3 based	35 (68.6)	26 (68.4)
CPX-351	15 (29.4)	3 (7.8)
HMA + Venetoclax	1 (1.9)	3 (7.8)
ICE-based	0 (0)	3 (7.8)
Other	0 (0)	3 (7.8)
Median duration of first remission	405 days	286 days
Prior salvage therapy	0 (0.0)	3 (7.9)
Molecular AML type		
t (8;21)	3/50 (6.0)	0/36 (0.0)
inv(16)	4/50 (8.0)	3/36 (8.3)
NPM1 mutated	8/50 (16.0)	11/35 (31.4)
FLT3-ITD	6/50 (12.0)	3/33 (9.1)
FLT3-TKD mutated	4/50 (8.0)	2/34 (5.9)
IDH1 mutated	7/48 (14.6)	0/22 (0.0)
IDH2 mutated	8/48 (16.6)	1/22 (4.5)
CEBPA biallelic mutation	2/47 (4.3)	1/22 (4.5)
CEBPA BZIP mutation	1/47 (2.1)	0/22 (0.0)
TP53 (VAF > 10%) mutated	11/37 (29.7)	2/20 (10.0)
ELN 2022		
Favorable	13/49 (26.5)	6/20 (30.0)
Intermediate	6/49 (12.2)	11/20 (55.0)
Adverse	31/49 (63.3)	3/20 (15.0)
Pre-existing conditions		
Cardiac conditions	13 (25.5)	5 (13.2)
Hepatic conditions	2 (3.9)	5 (13.5)
Renal conditions	2 (3.9)	2 (5.3)
Neurological conditions	5 (10.0)	3 (8.1)
Pulmonary conditions	7 (14.0)	10 (26.3)
Prior AHSCT	4 (7.8)	11 (28.9)

**Table 2 cancers-16-03872-t002:** Adverse Events in patients treated with FLA-VIDA vs. FLA-IDA regimen. Non-hematologic toxicity was evaluated according to the National Cancer Institute Common Terminology Criteria for Adverse Events (CTCAE v5.0). Hepatic injury was significantly less common in the FLA-VIDA group. Otherwise, there was no significant difference between treatment groups. Two grade 5 events occurred in the FLA-IDA group, one patient died due to subdural hemorrhage and one patient due to decompensated heart failure. No grade 5 events were documented in the FLA-VIDA group.

Adverse Events				
	FLA-VIDA		FLA-IDA	
	*n*	%	*n*	%
Total number of patients	51	100	38	100
Febrile neutropenia	45	88.2	35	92.1
Gram-positive bacteremia	13	25.5	15	39.5
Gram-negative bacteremia	3	5.9	5	13.2
Pneumonia				
≥CTCAE Grade 3	0	0	1	2.6
Any grade	13	25.5	17	44.7
Fungal pneumonia	3	5.9	3	7.9
Sepsis	7	13.7	5	13.2
Bleeding				
CTCAE Grade 5	0	0	1	2.6
≥CTCAE Grade 3	4	7.8	4	10.5
Any Grade	13	25.5	14	36.8
Tumor lysis syndrome (TLS)	6	11.8	3	7.9
Acute kidney injury				
KDIGO 1	5	9.8	5	13.2
KDIGO 2	2	3.9	4	10.5
KDIGO 3	1	2	4	10.5
Hepatic injury				
≥CTCAE Grade 3	1	2	6	15.8
Any grade	22	43.1	22	57.9
Heart failure				
CTCAE Grade 5	0	0	1	2.6
≥CTCAE Grade 3	3	5.9	0	0
Any grade	8	15.7	3	7.9
Diarrhea				
≥CTCAE Grade 3	2	3.9	2	5.3
Any grade	18	35.3	7	18.4
Mucositis				
≥CTCAE Grade 3	3	5.9	2	5.3
Any grade	14	27.5	7	18.4
Nausea				
≥CTCAE Grade 3	8	15.7	6	15.8
Any grade	19	37.3	10	26.3
Colitis				
≥CTCAE Grade 3	1	2	1	2.6
Any grade	6	11.8	3	7.9

## Data Availability

The datasets generated during and/or analyzed during the current study are available from the corresponding author on reasonable request.

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
