# Peer review of "Dose-Reduced FLA-IDA in Combination with Venetoclax Is an Effective and Safe Salvage Therapy in Relapsed and Refractory Acute Myeloid Leukemia (R/R AML)"

_cancers, 2024, doi:10.3390/cancers16223872_

Round 1
Reviewer 1 Report
Comments and Suggestions for Authors
The authors have conducted a single-centre, retrospective study of the efficacy and safety of salvage chemotherapy using venetoclax and lower dose FLA-IDA than previously reported and have shown results comparable to those previously reported.
The treatment of relapsed and refractory AML is an important clinical challenge to overcome and I believe this is a useful paper.
I would like you to consider adding or correcting information on the following points.
The background of the figure is black and the text is illegible.
When was NGS performed: at first presentation or at the start of salvage therapy?
How different is the follow-up time in the two groups?
Please indicate how many days after treatment the response was specifically assessed in both groups.
Were transplants performed after one course chemotherapy in both groups?
Please describe in detail what was done after one course in cases where no transplantation was done.
TP53 mutations are refractory and it is of interest how many TP53 mutation cases responded. Please describe how many responses were achieved in TP53 mutation cases.
There are two deaths in the FLA-VIDA group, but these are described as non-treatment-related deaths, what is the rationale for this?
Why were liver and kidney disorders more common in the FLA-VIDA group?
The discussion suggested that there was no difference in OS because some patients were saved by transplantation, but it is not appropriate to simply compare OS because of the different patient backgrounds.
The FLA-VIDA group may be more promising because of the high number of primary refractory and TP53 mutations. Discussion should be reconsidered.
FLA-IDA only compares cases where NGS was performed, which is a small number of cases, so why not compare it with a larger number of cases, including those where NGS was not performed? Comparisons with previous risk stratification by chromosome etc would also be helpful. Would there be a difference in OS if only primary refractory AML were subjected to matched-pair analysis?
In line 256 'Aes' should be corrected to 'AEs'.
Author Response
The background of the figure is black and the text is illegible.
We have included the revised graphics.
When was NGS performed: at first presentation or at the start of salvage therapy?
The NGS panel was conducted at the time of initial diagnosis. We have added a corresponding sentence in the ‘Material and methods’ section.
How different is the follow-up time in the two groups?
As the patients treated only with chemotherapy received their therapy before 2019, the follow-up period of the FLA-IDA cohort is notably longer. We have added a corresponding sentence in the results section.
Please indicate how many days after treatment the response was specifically assessed in both groups.
The days until remission control were added in the results section and are about 24.5 days each.
Were transplants performed after one course chemotherapy in both groups?
AHSCT was performed after one cycle of salvage therapy. We have added a corresponding sentence in the materials and methods section.
Please describe in detail what was done after one course in cases where no transplantation was done.
Here, too, we have added a corresponding sentence in the materials and methods section. Overall, an attempt was made to treat all patients with AHSCT even if they did not respond - if necessary, after further salvage therapy. Patients who did not receive AHSCT were mostly refractory to therapy with FLA-IDA/FLA-VIDA and subsequently not fit enough for further intensive chemotherapy or AHSCT.
TP53 mutations are refractory and it is of interest how many TP53 mutation cases responded. Please describe how many responses were achieved in TP53 mutation cases.
There are a total of 13 patients with a TP 53 mutation (2 in FLA-IDA and 11 in FLA-VIDA). In both cohorts the ORR was approx. 50%. Due to the small number of patients, especially in the FLA-IDA cohort, we decided against a comparison or more detailed analyses due to limited validity.
There are two deaths in the FLA-VIDA group, but these are described as non-treatment-related deaths, what is the rationale for this?
The patients died of an early and rapid progress of AML.
Why were liver and kidney disorders more common in the FLA-VIDA group?
Elevated liver enzymes and renal failure tended to be more frequent in the FLA-IDA cohort. We cannot give a certain explanation.
The discussion suggested that there was no difference in OS because some patients were saved by transplantation, but it is not appropriate to simply compare OS because of the different patient backgrounds.
If background refers to the genetic risk group, we can report that after further subdivision into the respective ELN risk groups, no reliable or statistically usable (OS) survival curves could be generated due to the small number of cases.
The FLA-VIDA group may be more promising because of the high number of primary refractory and TP53 mutations. Discussion should be reconsidered.
We dedicate a small paragraph in the discussion to this topic, which we expanded by one sentence in response to your feedback. Due to the small cohort groups, it is difficult to compare refractory AMLs or even TP53 mutations. However, it can be assumed that the advantage of the FLA-VIDA cohort might be even clearer with better balanced groups.
FLA-IDA only compares cases where NGS was performed, which is a small number of cases, so why not compare it with a larger number of cases, including those where NGS was not performed? Comparisons with previous risk stratification by chromosome etc would also be helpful. Would there be a difference in OS if only primary refractory AML were subjected to matched-pair analysis?
In an initial analysis, we categorized all patients according to the ELN risk classification valid at the time of initial diagnosis - thus all patients were assigned to the respective risk groups. There were no differences with regard to ORR, OS and EFS. In the primary refractory patients, the OS is not improved with FLA-VIDA, but the EFS is, which is reflected in the graphs for the entire FLA-IDA/FLA-VIDA cohort.
In line 256 'Aes' should be corrected to 'AEs'.
Corrected.
Reviewer 2 Report
Comments and Suggestions for Authors
The authors report dose-reduced FLA-IDA in combination with venetoclax is an effective and safe salvage therapy in relapsed refractory acute myeloid leukemia (R/R AML).
1. There is no information presented on pre-treatment of acute leukemia cases.
2. The number of eligible patients in the survival curves is not presented. Also, the background of the Figure is black, which needs to be improved.
3. The manuscript does not provide information on CPX-351.
4. The authors should describe the reason for the preference of FLA-IDA over FLA-VIDA in the ADVERSE arm of ELN2022.
Author Response
- There is no information presented on pre-treatment of acute leukemia cases.
We have added a corresponding paragraph in the material and methods section. There was only one first-line therapy in each case, most of them received 7+3. We have now mentioned the 18 patients that received CPX-351.
- The number of eligible patients in the survival curves is not presented. Also, the background of the Figure is black, which needs to be improved.
We have included the revised graphics.
- The manuscript does not provide information on CPX-351.
With 3 vs 15 treated patients in the respective cohorts, a comparison is not meaningful. We can report that a good ORR of 66.7% is achieved in the FLA-VIDA cohort despite the “more intensive” prior therapy with CPX-351. This is mentioned in the discussion.
- The authors should describe the reason for the preference of FLA-IDA over FLA-VIDA in the ADVERSE arm of ELN2022.
This can be explained by the small number of patients (n=3). There is a large confidence interval which clearly exceeds 1. We have included the adverse group for the sake of completeness, but the comparison with 3 patients has in our opinion no clinical relevance or statistical validity.
Round 2
Reviewer 2 Report
Comments and Suggestions for Authors
none